# A qualitative exploration of Australian eyecare professional perspectives on Age-Related Macular Degeneration (AMD) care

Isabelle Jalbert[1]*, Dian Rahardjo[1], Aryati Yashadhana[1], Gerald Liew[2], Bamini Gopinath[2]

**1** School of Optometry and Vision Science, UNSW Sydney, Sydney, Australia, **2** Centre for Vision Research, Department of Ophthalmology, The Westmead Institute for Medical Research, The University of Sydney, Sydney, Australia

* i.jalbert@unsw.edu.au

**Data Availability Statement:** All relevant data are within the manuscript and its supporting information files.

## Abstract

Despite the existence of evidence-based recommendations to decrease risk and progression of Age-Related Macular Degeneration (AMD) for some time, self-reported practices suggest that eyecare professionals' advice and people with AMD's adherence to these recommendations can be very poor. This study uses qualitative methods to explore Australian eyecare professionals' perspective on barriers to effective AMD care. Seven focus groups involving 65 optometrists were conducted by an experienced facilitator. A nominal group technique was used to identify, prioritize and semi-quantify barriers and enablers to AMD care. Participants individually ranked their perceived top five barriers and enablers with the most important granted a score of 5 and the least important a score of 1. For each barrier or enabler, the number of votes it received and its total score were recorded. Barriers and enablers selected by at least one participant in their top 5 were then qualitatively analysed, grouped using thematic analysis and total score calculated for each consolidated barrier or enabler. In-depth individual interviews were conducted with 10 ophthalmologists and 2 optometrists. Contributions were audio-recorded, transcribed verbatim and analysed with NVivo software. One hundred and sixty-nine barriers and 51 enablers to AMD care were identified in the focus groups. Of these, 102 barriers and 42 enablers were selected as one of their top 5 by at least one participant and further consolidated into 16 barriers and 10 enablers after thematic analysis. Factors impacting AMD care identified through analysis of the transcripts were coded to three categories of influence: patient-centered, practitioner-centered, and structural factors. Eyecare professionals considered poor care pathways, people with AMD's poor disease understanding / denial, and cost of care / lack of funding, as the most significant barriers to AMD care; they considered shared care model, access, and communication as the most significant enablers to good AMD care. These findings suggest that Australian eyecare professionals perceive that there is a need for improved patient support systems and appropriately funded, clearer care pathway to benefit people with AMD.

**Funding:** IJ, GL, BG received the Blackmores Macular Disease Foundation Australia research grant RG151916 (www.mdfoundation.com.au). The funders had no role in the study design, data collection and analysis, decision to publish, or preparation of the manuscript.

**Competing interests:** The authors have declared that no competing interests exist.

## Introduction

In 2013, 8.7% of the world's population aged above 45 years had age-related macular degeneration (AMD) and the number of people affected is projected to reach 196 million by 2020 and 288 million by 2040 [1]. In 2015–2016, one in six Australians over the age of 50 had AMD [2]. The impact of this common disease of the aging on people with AMD is significant. Contributing to approximately 70% of cases of bilateral blindness in non-indigenous Australians, AMD is the most common cause of blindness [2]. Having AMD (any stage of disease severity) has been shown to significantly impair activities of daily living [3]. Several modifiable risk factors have been shown to increase the risk of developing AMD and/or the speed at which the disease progresses [4, 5]. These include smoking and lifestyle risk factors such as dietary intake of antioxidants, low glycemic index diets, dark leafy green vegetables, fish and dairy product consumption [6–8]. Optimal AMD care can best be defined as appropriate and timely advice on uptake of dietary supplements and the importance of following a healthy diet and lifestyle (including refraining from smoking). For neovascular AMD, effective treatments that can slow the progression of disease are available in the form of anti-VEGF injections [6]. Evidence-based recommendations on appropriate modifications to recommend for people with and/or at risk of AMD have existed for some years [4, 9], yet self-reported practices suggest that eyecare professionals' advice to their patients on smoking cessation and dietary supplementation are often lacking [10–12]. In a survey of eyecare professionals in the UK, only one in three optometrists reported regularly assessing the smoking status of their new or existing patients and advising them on smoking cessation [10]. Although 93% of eyecare professionals appropriately recommended dietary supplementation to at-risk people, the type of supplements recommended often did not comply with current best research evidence, as suggested by the Age-related Eye Disease Study (AREDS) [10]; and there was a greater likelihood that ophthalmologists would recommend the AREDS formula than optometrists [10]. Similarly, a recent Australian survey indicated that fewer than half (47%) of optometrists routinely assess the smoking status of their patients [11]; factors such as optometrist's age and gender were associated with self-reported clinical practice behaviors, although no consistent pattern could be detected [11]. Comparative data on Australian ophthalmologists was unfortunately not collected.

Even when advice is appropriate, adherence to smoking cessation and dietary modification is often very poor among adults with AMD [13, 14]. Hospital based series have shown that people with AMD who have been recommended dietary supplementation are often not using them or are using an incorrect dose [14–16]. Conversely, one study showed that 20% of people with AMD who were using dietary supplementation did not have a level of AMD severity anticipated to benefit from that usage [15].

Barriers to the delivery of optimal eyecare result from a complex interaction of social, organisational, political, economic and cultural factors [17–19]. Little is known on what barriers specific to AMD may underpin the findings summarised above. Previous research has identified key barriers to the delivery of best practice eyecare. These include lack of time, lack of knowledge or skills, poor access to evidence, and inefficient care pathways [20–23]. In the setting of AMD, barriers such as poor awareness, professional development, socioeconomic disadvantage, cost, workforce supply, compliance and adherence issues have been postulated to exist [6, 24]. The lack of availability of evidence-based clinical care guidelines for AMD in many countries including Australia may also play a role [25, 26]. The current study aimed to explore the perspectives of eyecare professionals (optometrists, ophthalmologists) on AMD care in Australia, with a view to identify the most important barriers and enablers to appropriate AMD care and through this describe some of the possible reasons for under or over detection and inappropriate management.

## Materials and methods

The study received ethics approval number HC15776 from the Human Research Ethics Advisory Panel (HREAP) of the University of New South Wales (UNSW Sydney) in December 2015, in advance of the study start date. Written or oral consent was obtained from participants, depending whether interviews were conducted face to face or over the telephone. The study adhered to the tenants of the Declaration of Helsinki. A qualitative research approach was adopted to explore professional perspectives of eyecare for AMD. The study endeavoured to comply with the COnsolidated Criteria for REporting Qualitative Research (COREQ) [27]. The original intent was for this research to adopt a multimethod approach which combined several methods to collect information on a given topic [28]. A combination of focus groups (with the nominal group technique) [29] and semi-structured face to face or phone interviews were used to collect data from eyecare professionals. In this research, semi-structured interviews and focus groups were conducted in parallel and combined in a process known as method triangulation where there was mutual enhancement of the understanding of barriers and enablers to AMD care of each method by the other. Purposive, snowball, and maximum variation (e.g. urban versus rural location) sampling were planned to recruit eyecare professionals registered for practice as an ophthalmologist or an optometrist in Australia. Optometrists and ophthalmologists were recruited primarily from seven sites in three Australian states (New South Wales, Victoria, Queensland). In line with maximum variation sampling, these sites were selected to ensure the recruitment of eyecare professionals living in rural, urban and city locations. Recruitment continued until no new themes were emerging and data saturation had been achieved. Purposive sampling was applied to identify individuals across the age range, of both sexes, and those who were identified as specialising in the management of AMD as well as those from other specialty areas. To encourage participation of eyecare professionals, focus groups were held in the same location and at the same time as major national or state-based continuing education conferences (e.g. Australian Vision Convention 2016, Super Sunday 2016, Royal Australian and New Zealand College of Ophthalmologist Annual Scientific Congress 2016). All optometrists and ophthalmologists attending these conferences were invited to participate. Optometrists and ophthalmologists were also recruited by advertising in professional newsletters and email lists, through announcements at continuing education conferences, and by word of mouth. Early pilot work suggested probable difficulties in recruiting ophthalmologists to attend focus groups. Indeed, it was not possible to arrange enough participants (minimum of four) to conduct any ophthalmology focus groups. For that reason, the primary method of data collection used for optometrists was focus groups and for ophthalmologists was individual interviews. As a result, the final sampling method for this study is best described as convenience sampling, recognising that efforts were made to ensure appropriate representation where possible. Formal consent was obtained prior to or on the day of the focus group session or the interview, prior to commencing data collection.

Individual interviews were conducted with all ophthalmologists and two optometrists who wished to participate but could not attend the focus group sessions. An experienced eyecare professional, educator, and qualitative researcher (IJ) conducted all interviews. The length of interviews varied from 14 minutes to 33 minutes. The topic guide used during eyecare professional interviews was informed by a review of the literature and by issues raised during the focus groups. The semi-structured interviews and focus groups specifically aimed to identify and describe the most important barriers and enablers to optimal AMD care. The semi-structured interviews allowed for more detailed exploration of additional aspects to be undertaken. The topic guide covered the following key areas: experiences of AMD referral pathways, effectiveness of communication, compliance (health care professionals and patients) to evidence-

based care, sources of information, and opinions on barriers to appropriate care and how eye-care delivery for AMD could be improved (S1 Table). Follow-up probe questions were asked on an as-needed basis to facilitate the depth of the discussions. There was scope for participants to explore other relevant issues.

Focus groups lasted approximately 1.5 hours to 2 hours. The optometry focus groups were granted Continuing Professional Development (CPD) approval for 3 points in accordance with the guidelines from the Optometry Board of Australia. The focus groups were advertised under the Course Name "Breaking down barriers: What can be done to improve AMD care?" Participants received appropriate monetary compensation and light refreshments. One of two optometrists with public health training and experience in qualitative research (DR, Dr Nina Tahhan) co-facilitated the focus groups alongside an experienced optometrist, educator, and qualitative researcher (IJ) [20, 21]. The focus groups opened with a brief (5 minutes) presentation by the expert facilitator, summarising the current evidence-based classification, established risk factors, and management recommendations for AMD care. This included an overview of recent published evidence of eyecare professionals' advice and people with AMD's adherence to these recommendations being less than adequate. Seven focus groups were conducted in total. Barriers to AMD care were generated by all seven focus groups whilst enablers to AMD care were generated in a subset of three out of seven focus groups only. Background demographic information were collected on the participants prior to the commencement of each focus group. The nominal group technique was then used to elicit, prioritise and semi-quantify the participants' perspectives on the barriers and/or enablers to appropriate AMD care. The nominal group technique is a qualitative method of data collection that enables a group to generate and prioritise a large number of issues using a structure that gives everyone an equal voice [29]. It has been used in varying health contexts to generate ideas and allow groups to reach consensus on barriers and facilitators to health practices [30, 31] and was used for this research as previously described [21]. Briefly, participants in all focus groups were given an individual card and asked to record silently their responses to the question "From your perspective, what are some of the factors preventing people at risk of or with AMD from accessing and/or receiving good care and/or from following advice given to them?" In a subset of three focus groups (Gold Coast, Melbourne, Toowoomba), participants were also asked to record responses to the following second question "In your opinion, what more could be done to help people at risk of or with AMD to access and/or receive good care and/or to follow advice given to them?" Seating in the rooms in which the focus groups were held was configured in a U-shape. Participants took turns to read aloud a single response from their card with each response recorded on a flip chart at the front of the room. This continued in a round robin fashion until all responses were exhausted. A facilitated discussion occurred throughout this process to ensure that responses were reviewed and clarified, and a group consensus reached on the meaning of each individual contribution. This was followed by a group discussion where similar items were amalgamated. Participants were then asked to individually choose, rank, and record five responses they personally considered most important with the most important granted a score of 5 and the least important granted a score of 1. The rankings were summed for each item and the findings subsequently (after workshop) emailed back to participants for their information. For each barrier or enabler, the number of votes it received (maximum possible number is equal to the number of participants at each focus group) and its total (maximum possible score is equal to the number if a top rank of 5 was given by all participants in the focus group) and the number of individuals who ranked it first, second, third, fourth and fifth were recorded. All barriers and enablers were then reviewed independently by two researchers (DR, IJ) to identify and come to a consensus on common category themes generated across the different focus groups. The two researchers met to review participants'

responses, and this continued until a consensus had been reached. This iterative process allowed the category themes to emerge. The number of votes (total and by rank) and the total score for the barriers and enablers consolidated into a category theme were summed.

Focus group and interview contributions were audio-recorded, transcribed verbatim by a professional transcriptionist, and transcripts were checked by one of the researchers (IJ, DR) in preparation for qualitative data analysis. Analysis was conducted on transcribed interview data, by an experienced qualitative researcher (AY) using inductive thematic coding techniques and software NVivo 11 (QSR International, Melbourne, Australia) geared towards identifying patterns and discovering theoretical properties in the data. Interviews were coded 'openly' line by line drawing on grounded theory methods [32] which allowed a thematic structure to emerge organically from the data. This approach allows theory to evolve during actual research, and it does this through continuous interplay between analysis and data collection [33]. Thematic coding was conducted in a detailed manner, resulting in a large number of descriptive themes or 'codes'. This approach was taken so as not to lose context and enhance rigour in identifying saturation points. In addition to coding thematically, where applicable themes were also coded as a 'barrier' or 'enabler' to distinguish prevalence of these within eyecare professional perspectives.

The constant comparative method, a facet of grounded theory [32], was used to perform deeper analysis. The constant comparative method is described as a continuous growth process, where each stage after a time transforms itself into the next, and previous stages remain in operation throughout the analysis and provide continuous development to the following stage until the analysis is terminated [34]. In applying this method emergent themes were sorted by 'barrier' and 'enabler' and triangulated with the barriers and enablers identified during the nominal group process. Themes, including barriers and enablers, were discussed by two researchers (IJ, AY), merged and restructured, eventually resulting in three categories of influence (Table 1).

## Results

### Population

Seventy-seven eyecare professionals provided their perspectives of AMD care in Australia. Individual interviews were held with 10 ophthalmologists (7 face to face and 3 phone interviews) and two optometrists (face to face only). Sixty-five optometrists from across Australia also participated in seven focus groups (Table 2).

Three focus groups were held in Sydney (NSW) and one each in the Gold Coast (QLD), Melbourne (VIC), Toowoomba (QLD), and Orange (NSW). The number of optometrists participating in the focus groups ranged from 4 to 15. Two non-optometry AMD stakeholders (one low vision rehabilitation provider and one corporate communications representative of a professional eyecare journal and relative of a person with AMD) attended one workshop each. One provided no contributions to the focus group discussions. The low vision rehabilitation

**Table 1. Categories of influence within qualitative data.**

| Category of influence | Description |
|---|---|
| Structural | Factors related to health system or the processes within it such as referral pathways, human resources, equipment, funding. |
| Clinician-centered | Factors related to clinician's knowledge, experience, perspectives and practices. |
| Patient-centered | Factors related to patient's knowledge, experience, perspectives and practices. |

**Table 2. Characteristics and output of optometry focus groups.**

| Focus Group Location (Code) | No. of participants | No. of barriers | Barriers selected in top 5 | No. of enablers | Enablers selected in top 5 |
|---|---|---|---|---|---|
| Melbourne (FG1) | 15 | 20 | 17 | 21 | 17 |
| Gold Coast (FG2) | 11 | 19 | 10* | 14 | 13 |
| Sydney (FG3) | 15 | 38 | 23 | | |
| Sydney (FG4) | 4 | 27 | 14 | | |
| Sydney (FG5) | 10 | 28 | 14 | | |
| Toowoomba (FG6) | 5 | 19 | 13 | 16 | 12 |
| Orange (FG7) | 5 | 18 | 11 | | |
| **Total** | **65** | **169** | **102** | **51** | **42** |

* One participant ranked 2 items as 3 in importance. Barriers to AMD care were generated by all 7 focus groups whilst enablers were generated in a subset of 3 out of 7 only.

provider contributed to the Toowoomba focus group discussion, thereby providing additional data that is included in the sample and analysis.

The demographic characteristics of the participants are shown in Table 3. The Modified Monash Model (MMM) was used to classify eyecare professionals' geographical location, based on the postcode on their primary practice location [35]. All ophthalmologists practiced primarily in metropolitan regions (MM1-3). Five of 69 optometrists practiced primarily in a regional setting (MM4-5). No eyecare professionals participating in this study reported a primary place of practice located in rural or remote settings (MM6-7). Nine of 10 participating ophthalmologists worked in private practice in combination with other settings. Three ophthalmologists practiced in public hospitals and five in private day surgeries. Ophthalmologists reported providing a range of sub-specialty services including medical retina and AMD care (n = 8), cataract and anterior segment (n = 8), general ophthalmology (n = 6), vitreoretinal surgery (n = 3), glaucoma (n = 1), corneal and external diseases (n = 1), neuro-ophthalmology (n = 1), and paediatric ophthalmology (n = 1). Optometrists participating in the study practiced in a broad range of types of optometry practices. Fifteen worked primarily in an independent standalone private practice, 15 worked in educational facilities, 11 were part of a private practice buying group, nine worked in private practice franchises, five locumed in private practices, four worked in private practice corporations, three in community health services, three in other settings, and one each in hospital and in commercial / business services.

Eyecare professionals' perspective on factors impacting AMD care identified through analysis of the transcripts were coded into barriers and enablers and to patient-centered, structural, and clinician-centered factors, primarily.

## Overview

An overview of the semi-quantitative analysis conducted on the barriers and enablers generated during the optometry focus groups is presented initially before major factors (based on rank and cites) generated from both this analysis and the qualitative analysis of the transcripts

**Table 3. Demographic characteristics of the eyecare professionals participating in the study.**

| | Optometrists | Ophthalmologists |
|---|---|---|
| n | 67 | 10 |
| Age (years) | 41.4 ± 13.2 | 45.3 ± 7.2 |
| Gender (F:M) | 41:26 | 3:7 |

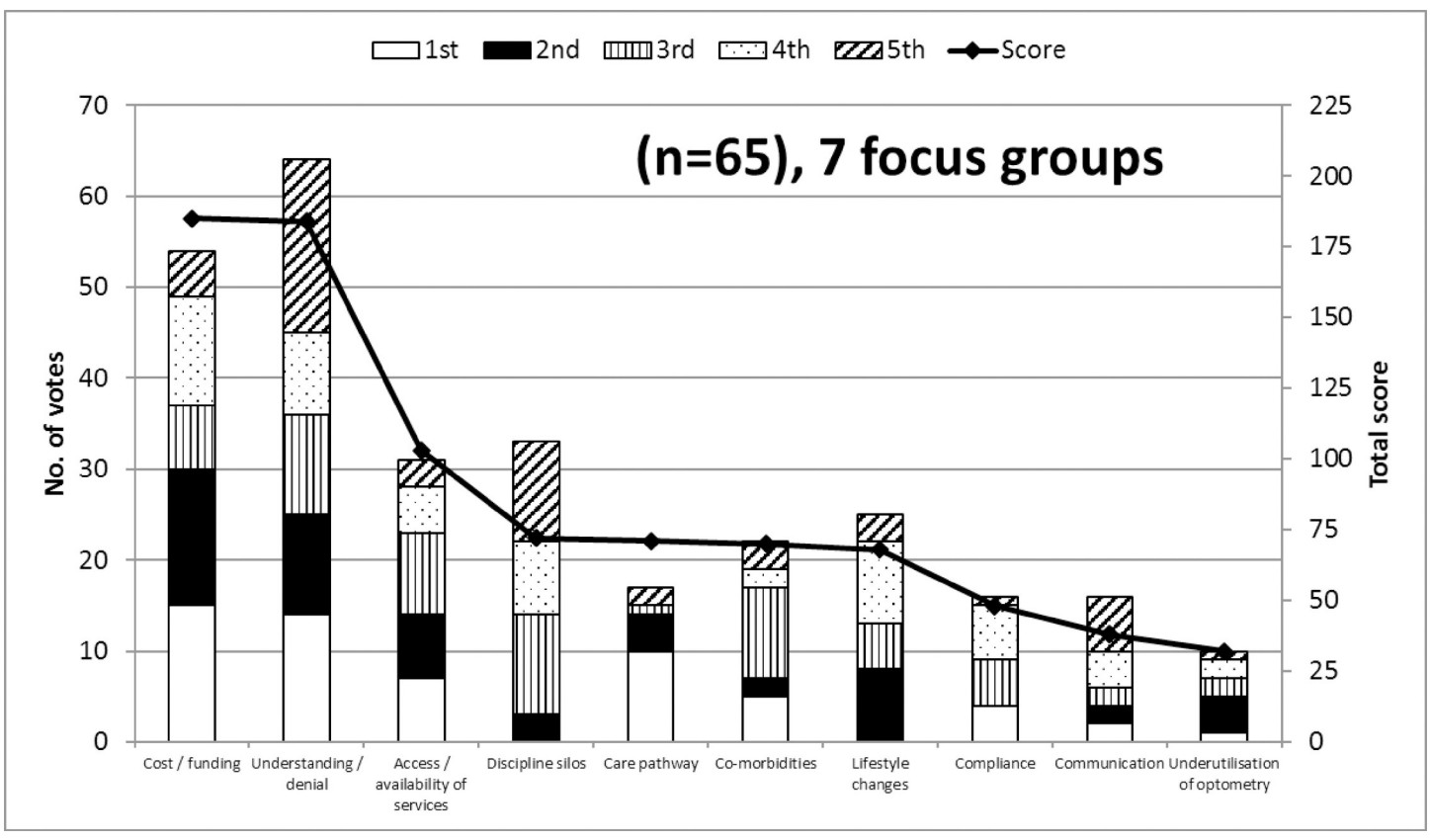

**Fig 1. Optometrists' perspectives on barriers to AMD care.** The most frequently cited (primary y-axis) and highest scoring (secondary y-axis) top 10 themes (representing barriers to AMD care) generated by the 7 optometry focus groups. The y-axis on the left is associated with the bars and represents the number of optometrists that identified these themes in their top five barriers and includes the detail of how many participants ranked each theme first (white), second (black), third (vertical lines), fourth (dots), and fifth (diagonal lines). The y-axis on the right is associated with the line and represents the top 10 total score given by focus group participants, based on how important optometrists felt these were (high score = high importance).

are developed. One hundred and sixty-nine barriers to AMD care were identified by the nominal group process in the seven focus groups attended by 65 optometrists. The characteristics and output in terms of number of barriers generated within each focus group are detailed in Table 2 above. Of these 169 barriers, 102 were selected as one of their top 5 by at least one optometrist. These 102 barriers were further segregated into 16 category themes after iterative analysis. Fig 1 illustrates the 10 most frequently cited themes that attracted the highest scores.

These represent the greatest impediments to AMD care as perceived by Australian optometrists. Although a substantial number of barriers were identified, the frequency at which the top two themes *cost/funding* and *understanding/denial* were cited was much greater than for all other themes or barriers. *Understanding/denial* was selected in the top five barriers by almost all participants (64 out of 65) and *cost/funding* was selected by 54 out of 65 participants. These two themes attracted the highest total score of 184 and 185, respectively, well above the next highest scoring theme *access/availability of services* at 103 (Fig 1). *Cost/funding* and *understanding/denial* were ranked as the top barrier (attracting a score of 5) by 15 and 14 optometrists, respectively.

Although well behind, other important themes consisted of *access/availability of services* (31 votes; total score 103), *discipline silos* (33 votes; total score 72), and *lifestyle changes* (25 votes; total score 68). S2 Table presents the AMD care category themes and barriers to AMD care that were selected in the top five by at least one optometrist in at least one focus group.

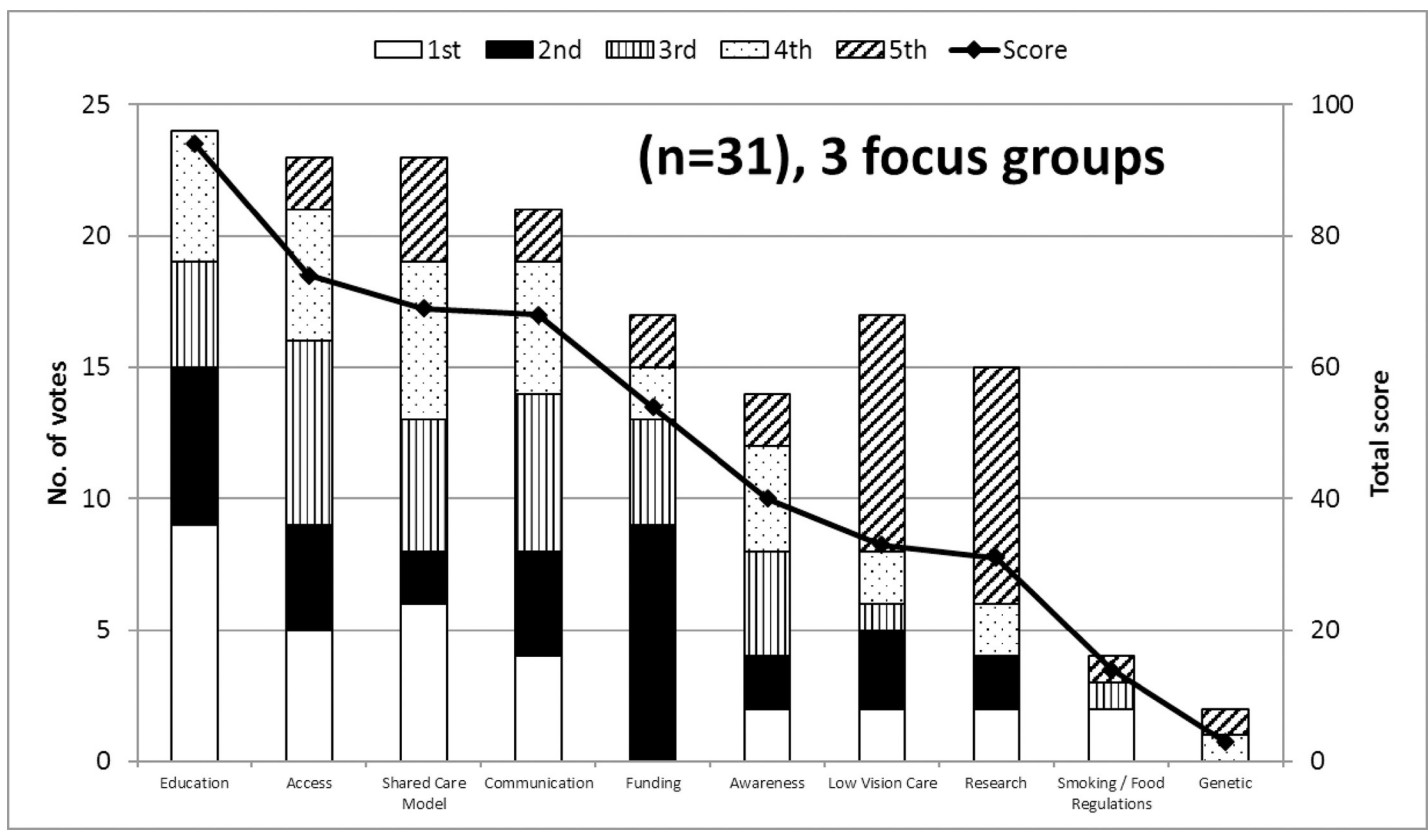

**Fig 2. Optometrists' perspectives on enablers of AMD care.** The most frequently cited (primary y-axis) and highest scoring (secondary y-axis) top 10 themes (representing enablers of AMD care) generated by the 3 optometry focus groups. The y-axis on the left is associated with the bars and represents the number of optometrists that identified these themes in their top five enablers and includes the detail of how many participants ranked each theme first (white), second (black), third (vertical lines), fourth (dots), and fifth (diagonal lines). The y-axis on the right is associated with the line and represents the top 10 total score given by focus group participants, based on how important optometrists felt these were (high score = high importance).

S3 Table presents the list of 67 barriers that were identified by optometrists during any one of the focus groups but were not selected during "top five barriers" ranking process for that group (note that a number of these barriers may have been ranked by alternate focus groups). Most of these 67 barriers could be segregated in the 16 category themes identified in the iterative analysis (S3 Table). Two additional category themes were identified, those being *Amsler grid* and *supplements*. Two barriers could not be categorised in any of the 18 category themes and are listed in S3 Table as miscellaneous.

Fifty-one enablers of appropriate AMD care were identified by the nominal group process in the subset of three focus groups attended by 31 optometrists. The characteristics and output in terms of number of enablers generated within each focus group are detailed in Table 2 above. Of these 51 enablers, 42 were selected as one of their top 5 by at least one optometrist. These 42 enablers were further segregated into 10 category themes after iterative analysis and these are illustrated in Fig 2.

These represent the greatest enablers of good AMD care as perceived by Australian optometrists. *Education* was the most frequently cited enabler, being cited by 24 of 31 participants and it attracted the highest score of 94, more than 20 points above the next highest scoring enabler. *Access* (23 votes; total score 74), *shared care model* (23 votes; total score 69), and *communication* (21 votes; total score 21) were also considered important themes. S4 Table presents the AMD care category themes and enablers to AMD care that were selected in the top five by

at least one optometrist in at least one focus group. S5 Table presents the list of nine enablers that were identified by optometrists during any one of the focus groups but were not selected during "top five enablers" ranking process for that group.

The amalgamation of barriers involved much discussion by participants during the focus group consensus building exercise as well as by researchers involved in the iterative thematic analysis. Where full agreement could not always be reached, a compromise position may have been adopted. As a result, readers may perceive similarities between barriers that were not amalgamated and distinctions between those that were. Areas of potential cross-over are discussed in more details here. The theme *understanding/denial* is a broad theme that was amalgamated from 16 identical and/or related barriers identified and described by focus group participants. These barriers encompassed concepts related to denial and fear (including fear of injection), mental state, lack of understanding, complacency, and lack of knowledge. Central to this theme was the notion that AMD is often asymptomatic at time of diagnosis. Many discussions during focus groups centered around the consequences of this lack of symptoms on people with AMD's understanding and belief regarding their disease and how this could combine with fear to enable denial. The poor care pathways theme centered around the complexity, and lack of clarity of the journey back and forth to and between services and providers for people with AMD. This went beyond optometry or ophthalmology to include general practitioners, dieticians, low vision and rehabilitation services providers and even included discussion of transport subsidies and care managers. The underutilisation of optometry theme although related centered around the role of optometry as a primary care provider that can screen, detect, triage and direct people with AMD to relevant services. The care guidelines theme focused not on the journey between service providers back rather on the lack of availability of evidence-based clinical care guidelines and the resulting lack of clarity as to what the best or most appropriate treatment might be for people with AMD."

Qualitative analysis allowed coding of the transcripts from the optometry focus group and individual interviews into themes as well as into barriers and enablers that were then categorised into one or more of three categories of influence: structural, clinician-centered and patient-centered (Table 1, Fig 3). Individual themes could be coded as barrier or enabler or both. Key themes are developed and described below, with emphasis on those themes that overlapped more than one category of influence. Supporting example quotes are provided. When responses of focus groups and interviews where participants discussed both barriers and enablers to AMD care were compared to those focus groups where participants discussed barriers only, no clear patterns or differences could be detected.

## Communication and trust

The themes of trust and communication sat at the intersection between the patient-centered and the clinician-centered categories of influence. Communication was considered an important enabler by eyecare professionals (Fig 2). Lack of trust was often associated with people with AMD's lack of understanding and this was perceived to be related to the complexity of the information that is given to them.

"*My father in law, he's suffering now, he can't be bothered going, [. . .] doesn't understand why he's getting his injections and he's [. . ..] to go and see a local ophthalmologist.*" Optometrist, FG2

"*I think telling people, eat more fruit and vegetables, low GI food and some nuts, it just sounds like every bit of dietary advice, and it doesn't sound special, and they've got a special eye threatening disease, and that just sounds so generic, like how can that possibly work. There's*

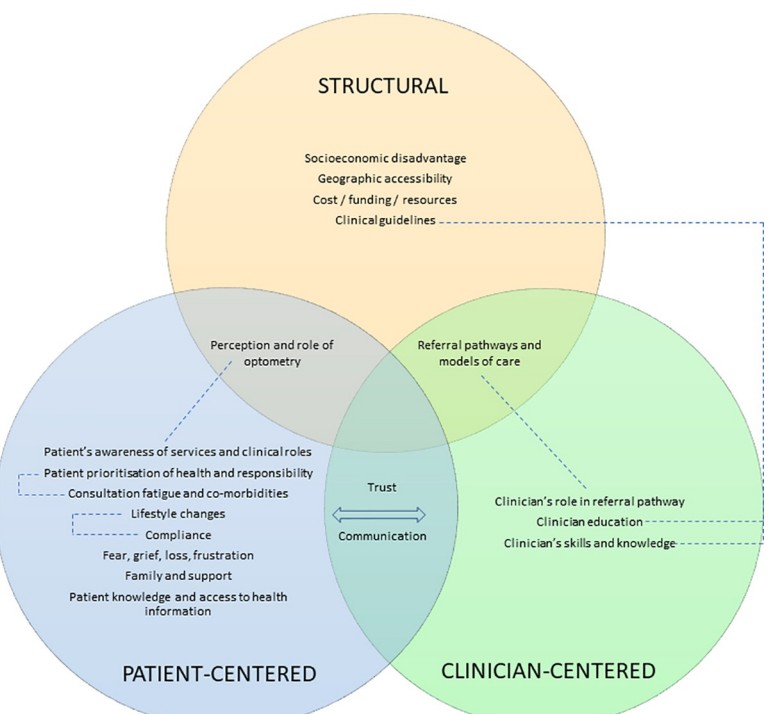

**Fig 3. Overview of three categories of influence in a Venn diagram: Structural, patient-centered, and clinician-centered themes generated by the qualitative analysis and their overlap and relationships.** Themes were categorised into structural (yellow circle), patient-centered (blue circle), and clinician-centered (green circle) categories of influence. Those themes that overlapped more than one category of influence such as for example trust and communication were placed at the appropriate intersection of the Venn diagram. Significant relationships between themes are highlighted with dotted lines. Note that any theme could be classified as a barrier, enabler, or both, depending on the eyecare professional's individual context.

*distrust of it working and there's a feeling that they've heard it all before and it doesn't do anything.*" Optometrist, FG3

Optometrists and ophthalmologists felt that poor communication with people with AMD could represent a significant barrier to appropriate care and that this could occur both at the primary (optometry) and secondary (ophthalmology) AMD care stage.

"*[. . .] it's only going to be possible for us to manage what we're doing in public hospitals by having optometrists dealing with early stages of macular degeneration and being able to explain to the patients well and not cause them worry. I think I deal often with people who are unnecessarily worried just because [of] the words [that] have been used [by the optometrist].*" Ophthalmologist P54

"*He [the ophthalmologist] never explained anything.*" Optometrist, FG7

A lack of shared decision making and communication in AMD care and poor involvement of the people with AMD in decisions regarding their care plans by eyecare professionals was also identified as a problem.

"*I get a lot of clients who come and say, they told me I have to do this, I don't know why, I just, they told me I have to, but I don't want to, 'cos he said I have to! So it's involving*

*the person in that decision making for their own health and wellbeing maybe.*" Optometrist, FG6

## Referral pathways and models of care, perception and awareness of clinical roles

At the intersection of structural, clinician- and patient-centered categories of influence sits the themes related to referral pathways for AMD care and the understanding and perception of the various eyecare professionals' roles in the care model for AMD. These were perceived by eyecare professionals as significant barriers but at the same time as key enablers to AMD care. This is perhaps a reflection of the significance of these factors, whether they are well or poorly articulated and implemented in the Australian healthcare system. A number of the themes considered distinct by optometrists participating in the focus group sessions (see Fig 1) are encompassed in this factor including *discipline silos*, *care pathway*, and *underutilisation of optometry*.

Optometrists observed that they were often underutilised and recounted incidences of missed opportunities where their profession could have helped to provide care for people with AMD, but these people with AMD were instead directed to secondary or tertiary care by their general practitioner or relatives and friends. This factor overlapped with that of the perception of the role of optometry, from the perspective of people with AMD and other healthcare professionals.

"*One of the things [. . .] is practitioner cognition of the optometry's role which comes down to professional understanding issues, people are seeing their GP [general practitioner] and the GPs are referring to ophthalmology.*" Optometrist, FG2

Poor patient awareness of services and clinical roles was cited, as well as poorly developed referral pathways and models of care.

"*For instance, we have diabetic patients, for diabetic patients we have a diabetic plan and the person has control of it, and that GP goes, you need to see your optometrist or ophthalmologist, you need to see a nutritionist, you need to see a podiatrist, I need to see you back in three months, and every three months he goes over it again, and goes over it again, and makes sure [. . ..] And for people with macular degeneration it is kind of like, [. . .] there is a lot of information but there is nothing that puts it all together in a proper flow chart for anybody to find and it's confusing for patients.*" Optometrist, FG2

"*Along the side of the barriers, is, for example, if optometrists actually refer to dieticians and are they allowed to or do they have to go through a GP to get a referral?*" Optometrist, FG4

Some eye health professionals also felt that the scope of services offered may vary between practice types and that this may impact the effectiveness of the care for people with AMD and their care pathway.

"*So if we're expecting people to refer to optometrists to diagnose and manage macular degeneration patients, how would anyone know which optometrist is appropriate to refer to when you could have anything from 10 minute refraction through to the whole gamut bordering on the level of investigation of an ophthalmologist, there's a huge scope in the way we practice.*" Optometrist, FG2

*"Again, it probably reflects dealing with the public hospital workload, it's only going to be possible for us to manage what we're doing in public hospitals by having optometrists dealing with early stages of macular degeneration and being able to explain to the patients well and not cause them worry."* Ophthalmologist P54

## Cost, funding and resources

This factor explored the extent to which cost impacted the ability for AMD care to be delivered and received in Australia. The structural barriers, cost and inadequate funding, were cited frequently by eyecare professionals as a significant barrier to good AMD care. This was consistently expressed across the various focus groups and included cost of transport, the out of pocket consultation costs faced by people with AMD when consulting various eyecare professionals (optometrists, ophthalmologists), optometry's access to equipment (Amsler grid, optical coherence tomography, autofluorescence), the cost of ancillary tests (e.g. fundus photos, optical coherence tomography scans), the cost of accessing other types of services (e.g. dietician, counselling), and the cost of various available management and treatment strategies including dietary supplements, healthy foods, anti-VEGF injections and low vision aids, and the inadequacy of government support provided through Medicare funding and the Pharmaceuticals Benefit Scheme, and the costs associated with publicly funded versus private health insurance care.

*"Lack of funding for various aspects of care (e.g. counselling); patients take up lots of chair time but not profitable–barrier to good care."* Optometrist, FG2

*"And then finally with the health system I've also talked about the time consultation with Medicare, you've got a full book, it's all very well to say you should counsel that person on quitting smoking and you should counsel that person on the size type of fish they should eat, and the colour vegetables that are best for them, in a 15 minute consult time, which you need to be able to pay your bills, you can't do it. So there is that, I guess the reimbursement for the time that you're spending with the patient, it's inadequate. We don't get paid enough. It should be $120 a visit, it's as simple as that, it's garbage."* Optometrist, FG3

*"We need subsidies for low vision aids."* Optometrist, FG2

*"Not many ophthalmologists bulk bill, in fact, hardly anybody where I am."* Optometrist, FG3

*"If you've got dry macular disease and you want just public care then I think that's something you're not going to get because the hospitals just don't have the capacity to see those patients every twelve or twenty-four months, if there is not additional need or intervention."* Ophthalmologist P60

*"Still poor access to autofluorescence and OCT in primary care optometry."* Optometrist, FG3

## Clinician's knowledge and skills

Participants listed eyecare professionals' knowledge and skills and their ability to properly diagnose and treat AMD as an important clinician-centered factor. As discussed above, ophthalmologists were generally supportive of optometry's role in the care pathway but there were occasional reservations expressed regarding optometry's skills in AMD care. This intersected with the availability of specialised equipment discussed above.

*"I get a lot of referrals from solo practitioners who've got no cameras or any, and they seem to have very little knowledge or understanding about or ability to distinguish pathology at the back of the eye. I mean, they're relatively a minority but there's quite a lot around."* Ophthalmologist P62

For ophthalmology, it was felt that the potential earnings from AMD care, particularly in the context of government supported anti-VEGF injections, may attract ophthalmologists to expand their services into this area without perhaps the necessary expertise.

*"I do think the lure of AMD treatments attracts many people with varying degrees of expertise to participate in the care of AMD patients. So look through the community, there are a lot of people calling themselves retinal specialists who in fact have no particular training in retina. So I think that, whereas in the past they have had zero interest in having that patient under their care."* Ophthalmologist P56

*"There's a motivator for that patient to stay in that practice, ok, intravitreal injection, you don't need to be that skilled to do it, in some ways, but you do need to be very skilled in making a diagnosis and monitoring the patient and getting the right balance between treating them appropriately and not treating them. So I would say that, and I do some regional work as well, cost can be a barrier in regional centres where there is limited public hospital access and there are, you know, potentially ophthalmologists in a limited number, who have significant fee structures that are perhaps even higher than what would be seen in the city."* Ophthalmologist P55

A number of optometrists also expressed some uncertainty regarding their knowledge whilst at the same time expressing a desire for more targeted education.

*"We hear all these professionals at conferences and whatever, most of it's all sort of on that really high level, but I don't think most of us really understand, really what we should be telling people [at risk or with macular degeneration]. You said yourself that only half of us asked whether our patients are smoking, [referring to the talk prior to focus group] you know, why is that? So we need kind of this simple education, this is where we what, really, the science is saying about supplements, this is what the science is saying about diet, this is–I reckon we're all a bit confused as well out there in working land."* Optometrist, FG6

## Access (geographical, availability)

The structural factor *access* intersects with the referral pathways above and was perceived by eyecare professionals as both a significant barrier and an important enabler to good AMD care. Eyecare professionals expressed the view that people with AMD were often highly dependent on having the appropriate support to enable them to access the care they required. This included people in the form of families and friends but also systems such as supportive nursing home staff and adequate transport systems (e.g. bus).

*"So mine was more about the lack of support because older people are more dependent on their family and friends and they don't see a doctor independently. Like the older people have given up their licences because they feel they're not safe enough to drive or anything like that so it's hard for them to get to these places even if we refer them to, say, Vision Australia or you refer them to other health care providers. It's hard for them to get around when they can get access to them."* Optometrist, FG3

The need for more frequent visits associated with treatments requiring regular injections (e.g. every 4 to 6 weeks) was frequently cited as a specific factor that significantly complicates access to AMD care.

> "*I think, particularly in the country, transport's a big issue, to actually try and get to see an ophthalmologist, they're living probably on their own and maybe riding a scooter around and they can't get into town. They might be able to come once or twice but then for treatment, for injections and things, it's just difficult.*" Optometrist, FG6

## Discussion

Our findings suggest that a combination of patient (e.g. trust), clinician (e.g. models of care) and structural (e.g. health care system cost) factors present significant barriers to AMD care. Australian eyecare professionals perceive that patient support systems and appropriately funded, clearer models of care would benefit people with AMD.

### Models of care: A role for AMD case managers?

Many optometrists and ophthalmologists commented on the lack of effective AMD care coordination in the current health system. Eyecare professionals' descriptions of a unidirectional, ineffective model of AMD care aligns with previous research findings [36–38]. Previous research has suggested that the role of the optometrist in the ophthalmic care pathway often goes unrecognised, with optometrists seen differently from other healthcare professionals [39]. In line with the findings of this study, a combination of service/healthcare professional-related factors (e.g. inability to get an early appointment, AMD not appropriately detected or labelled) and patient-related factors (e.g. lack of knowledge of AMD symptoms and/or risks) have been cited as common reasons for delay in diagnosis of AMD [40]. Many people with AMD referred for ophthalmological care do not require specialised services or ongoing treatment in a secondary or tertiary setting [38] and could benefit from better recognition and usage of primary eyecare services. The reasons underpinning this are likely complex and may be related to the public perception of the optometry profession. For example, adults surveyed in the United Kingdom believe that spectacles are overpriced, they perceive sales tactics to be dubious and that optometrists cannot be trusted to give impartial advice [39].

Focus group and interview participants in this study frequently contrasted these known findings with the perceived effectiveness of the diabetes care model in Australia, where care is supported through appropriately funded diabetic care plans, managed by general practitioners acting as case managers. Participants felt that the diabetes care model ensured better continuity, access, and timeliness of eyecare in Australia. Case management provided by workers such as nurses or community health workers have been shown to improve diabetic care outcomes [41]. To the best of our knowledge, the potential effectiveness of having case managers and/or community health workers provide intensive support to people with AMD has not been formally tested. A "fast track" referral pathways where people with wet macular degeneration are referred from the optometrist or the general practitioner directly to a specialist AMD center for triage has been advocated in the United Kingdom [42]. Should similar recommendations be adopted in Australia as well as in the United Kingdom, AMD case managers could provide effective liaison between the people with AMD and the healthcare and aged care systems. In line with evidence and the findings from this study, AMD case managers could ensure adequate and timely access to services focusing on education delivery and smoking cessation, nutrition, physical activity, orientation and mobility, low vision rehabilitation, driving support

and cessation, medication and dietary supplementation adherence, appointment adherence, psychological counselling, and socioeconomic issues. In an Australian context, the low population density and cost of case managers are factors to be considered.

## Clinician's skill and knowledge: Accuracy of diagnosis and treatment advice

Clinicians' skills and knowledge were identified as significant barriers to good AMD care in this study. In support of this, a recent cross-sectional audit study of people examined in primary eyecare practices in the United States and deemed normal revealed that approximately 25% of eyes had AMD based on fundus photography assessment by trained raters [43]. In the same study, 30% of undiagnosed eyes had AMD with large drusen that would have benefited from a recommendation to use nutritional supplements [43]. Ophthalmologists and optometrists in these primary eyecare practices were equally likely to misdiagnose AMD [43]. Between country differences may complicate this issue. In an analysis of quality of 54 rapid access referrals to a neovascular AMD clinic in the United Kingdom, optometrists' ability to accurately recognise common signs of macular degeneration such as drusen and sub-retinal fluid was low at 52% and 44%, respectively [44]. The overall accuracy of optometry diagnosis was low as 63% of referrals to this clinic did not have macular degeneration [44]. Conversely, in a retrospective review of patients records and referrals for macular disease at a diagnostic imaging centre in Australia, excellent diagnostic congruency between the optometric referral and the centre was observed with 47 of 50 or 94% of cases correctly diagnosed by the referring optometrist [45]. A lack of trust between optometrists and ophthalmologists in Australia was identified as a fundamental issue to address [38]. This issue sits at the intersection between clinician's skills and knowledge and models of care, with this lack of trust likely to hinder any possible progress in enacting any new proposed models of care. In a qualitative study of a shared care model in Australia, ophthalmologists expressed reservations about the skills of optometrists [38] and this was reflected at times by opinions expressed in this study. UK healthcare professionals when asked for their perspectives on care for wet or neovascular macular degeneration also spoke about optometry competency and models of care as important factors or concerns [46]. This issue remains an ongoing impediment to progress on shared care for AMD.

This study also found that there remains confusion among eyecare professionals, most particularly optometrists, about what supplements and what foods should be recommended for people with AMD. Previous data from surveys of optometrists in the UK and Australia has highlighted deficiencies in eyecare delivery in the areas of risk factor assessment, dietary supplementation advice, and diagnostic techniques.[10, 11, 47]

## Education

Education was ranked as the top strategy to improve AMD care delivery by optometrists in this study (Fig 2). This theme included education of eyecare providers, people with AMD and the general public (S4 Table) but the focus groups did not provide any insights on the potential design, feasibility and effectiveness of these educational interventions. One such example might be a clinical decision-making tool in the form of a flowchart, which was recently shown to aid qualified and student optometrists in the UK match patients to the AREDS 2 eligibility criteria [48]. The "nutrition advice for people with, or at risk of AMD" clinical decision-making tool was shown to increase practitioner confidence and lead to improved student optometrist performance on five hypothetical clinical scenarios [48].

Group-based health education programmes for people with AMD has previously been shown to be beneficial, notably through the provision of social support and through meeting other people

with AMD [49]. Unfortunately, problems with understanding the information provided by the eyecare professionals were still experienced and reported in attendees of these health education programmes [49]. Educational interventions based on focus group findings have in the past been unsuccessful in other fields when these failed to properly consider the complexity of the recommended management and when the intervention failed to include a combination of different approaches [50]. Shaw and colleagues have unfortunately shown that patient education by optometrists in the United Kingdom was likely to be underutilised and inconsistent: very few optometrists in their study discussed glaucoma risk factors with a patient of African racial decent, even when the standardised patient asked the optometrists if she was at greater risk of any eye conditions [51]. Sustained interventions are likely to be required in order to effect long-term behaviour changes in eyecare professionals and in people with AMD. This will require significant policy commitment and dedicated resources from the Australian government.

## Communication

Repeated surveys of Macular Disease Society members in the UK highlight significant dissatisfaction with their consultation in people with AMD with 24.2% of people with AMD dissatisfied with their consultation in 2013 [52, 53]. The most common reason for dissatisfaction with diagnostic consultation was remains poor specialist's attitudes (often described as dismissive, patronising, brusque, unfeeling, uninterested, etc. - 47.8%) [52, 53]. Over half of people with AMD indicated that they do not receive enough information about nutrition and lifestyle choices [54]. Many studies have highlighted the negative impact of an AMD diagnosis on individuals' emotional wellbeing and the inability of eyecare professionals to provide the type of information and support that is required by people with AMD [55, 56]. Ninety percent of participants reported having been told "nothing can be done about your macular disease" by their specialist and reported feeling anxiety / depression, resignation, shock / sick/ panic, helplessness, anger, and suicidal as a result [52, 56]. These findings highlight the importance of effective communication between eyecare professionals and people with AMD. In the current study, communication and trust were placed at the intersection of patient-centered and practitioner-centered factors. Public perception of optometrist as retail businesspersons with little to no healthcare role may affect the professions' credibility and the people's trust in the perceived utility of optometrist recommendations [57].

## Strengths and limitations

Strengths of this study include the mixed methods approach that allowed triangulation of the data and enabled the participants to express concerns in their own words, the involvement of primary and secondary eyecare professionals, and the use of nominal group technique which allowed quantification of barriers and enablers. The concurrent use of an explorative and consensus-oriented method allowed both qualitative and semi-quantitative data to be collected, inform and complement each other but the lack of ophthalmology involvement in the focus group aspect of this research is a limitation [28]. These strategies are known to be helpful when trying to understand complex behaviours and characterise knowledge, attitudes, and barriers of diverse stakeholder groups [30, 58, 59]. The number of barriers identified by different focus groups varied from 18 to 38 (Table 2). This variability perhaps reflects the characteristics inherent to focus groups conducted with the nominal group technique: those where a low number of barriers was identified may have involved a dominant participant whereas those focus groups where a higher number of barriers were generated may have involved participants with more diverse views. The level of homogeneity present in the participants involved in each focus group (e.g. age, gender) may also have influenced these findings.

There are some potential limitations to this study. The brief presentation at the opening of the focus group may have influenced participants responses. Less than half of the focus groups (three of seven) asked about enablers; the interview topic guide was also heavily weighted towards barriers. This is likely to have biased the results towards a focus on barriers. The number of ophthalmologists interviewed for this study was small and they were all from New South Wales and thus, the findings on the ophthalmology perspective are based on fewer comments and may not be representative of the whole of Australia. These should be interpreted with caution as state-based factors from other Australian states may have been missed. Recruitment difficulties prevented the conduct of planned ophthalmology focus groups; the semi-quantitative findings presented in this study may not represent the ophthalmology perspective. Participants were also relatively young, in their 40s and 50s, and may not be reflective of views and perspectives of older eyecare professionals. The generalisability of these findings outside of Australia is also unknown. The findings of this research may not be unique to the setting of macular degeneration or the Australian context. Selection bias may have been present in this study. Eyecare professionals who volunteered to participate in the study may have been those who have better knowledge and familiarity and a more positive attitude towards AMD care. However, the effect of this would be to underplay the barriers to AMD care. The findings reported in this study represent the perspective of eyecare professionals on a research question that also involves people with AMD and their access to care; eyecare professionals' perspective of the motives and behaviour of people with AMD should be interpreted with caution. This study's findings would therefore benefit from cross-tabulation with the perspective of people with AMD and their carers. This research has been conducted and will be reported separately.

Significant research has been conducted to understand the barriers to use of clinical practice guidelines and evidence-based practice by health professionals [18, 60, 61] and/or barriers to patients' access to care [19]. From this research a number of theories, models and frameworks including the well-known theoretical domains framework (TDF) [60] are available that can be applied to research such as that presented in the current study to help understand, explain, and address the gaps between research and practice in the area of AMD care. Following the identification of the determinants of AMD care (i.e. barriers and enablers) through this study, application of one or several of these frameworks to the study findings could help to refine and confirm the appropriateness of the recommendations suggested below. This work is currently underway and will be reported separately.

## Recommendations

The results of these focus groups and interviews provide guidance on the kinds of intervention that can improve the appropriateness of AMD healthcare delivery in Australia. On the basis of the findings from this study, the following initial recommendations are made:

- Clear, effective models of care should be developed collaboratively (with the involvement of all stakeholders), tested, and adequately incentivised

- Research into the potential benefits and cost-effectiveness of public funding support of case managers for people with AMD is warranted

- Affordable, accessible transportation services for people with AMD need to be broadly available

- Appropriately designed (e.g. group-based programs, decision aids, etc.) educational interventions directed at people with AMD, ophthalmologists, and optometrists could be beneficial

A comprehensive approach that encompasses clinical training, research into the most appropriate models of care, followed by the dissemination of clear evidence-based referral guidelines and pathways with appropriate associated remuneration (e.g. Medicare) is required.

## Supporting information

**S1 Table. Semi-structured Interview Guide.**
(DOC)

**S2 Table. Ranked Category Themes and Barriers to AMD Care.** AMD care category themes and barriers to AMD care, category of influence and rank-ordering votes across the seven optometry focus groups.
(DOCX)

**S3 Table. Additional Unranked Barriers to AMD Care.** Barriers nominated by optometrists in at least one focus group but not selected during the "top five barriers" ranking process.
(DOCX)

**S4 Table. Ranked Category Themes and Enablers to AMD Care.** AMD care category themes and enablers to AMD care, category of influence and rank-ordering votes across three optometry focus groups.
(DOCX)

**S5 Table. Additional Unranked Enablers.** Enablers nominated by optometrists in at least one focus group but not selected during the "top five barriers" ranking process.
(DOCX)

## Acknowledgments

We thank all the optometrists, ophthalmologists, and stakeholders who participated in the study. The state branches of Optometry Australia Optometry New South Wales/ACT, Optometry Victoria, and Optometry Queensland/Northern Territory provided in-kind and logistical support towards the conduct of the focus groups. All recordings were transcribed by Ms Sylvia Madden, Paekakariki, New Zealand. Dr Nina Tahhan is acknowledged for her generous contribution towards the moderation of 3 of the 7 focus group sessions.

## Author Contributions

**Conceptualization:** Isabelle Jalbert, Gerald Liew, Bamini Gopinath.

**Data curation:** Isabelle Jalbert, Dian Rahardjo, Aryati Yashadhana, Bamini Gopinath.

**Formal analysis:** Isabelle Jalbert, Dian Rahardjo, Aryati Yashadhana, Bamini Gopinath.

**Funding acquisition:** Isabelle Jalbert, Gerald Liew, Bamini Gopinath.

**Investigation:** Isabelle Jalbert, Dian Rahardjo, Gerald Liew.

**Methodology:** Isabelle Jalbert, Dian Rahardjo, Aryati Yashadhana, Gerald Liew, Bamini Gopinath.

**Project administration:** Isabelle Jalbert.

**Resources:** Isabelle Jalbert, Gerald Liew.

**Supervision:** Isabelle Jalbert.

**Validation:** Isabelle Jalbert.

**Writing – original draft:** Isabelle Jalbert, Aryati Yashadhana, Gerald Liew, Bamini Gopinath.

**Writing – review & editing:** Isabelle Jalbert, Dian Rahardjo, Aryati Yashadhana, Gerald Liew, Bamini Gopinath.

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
