## [Decision Letter · Decision Letter 0]

10 Jan 2020

PONE-D-19-30557

A qualitative exploration of Australian eyecare professional perspectives on Age-Related Macular Degeneration (AMD) care

PLOS ONE

Dear A/Prof Jalbert,

Thank you for submitting your manuscript to PLOS ONE. After careful consideration, we feel that it has merit but does not fully meet PLOS ONE’s publication criteria as it currently stands. Therefore, we invite you to submit a revised version of the manuscript that addresses the points raised during the review process.

ACADEMIC EDITOR: 

Thank you for this manuscript re-submission, which has been considered as a new submission. The study provides a thoughtful consideration of  aspects of current management and care of patients with Age-related Macular Degeneration in Australia.

1. The reviewer raises several issues, in particular the observation that the focus guides and questions do seem to be more directed at 'barriers' than 'enablers'. This  potential bias in study design should be addressed within the Discussion. 

2. As noted by the reviewer, indicating whether identified barriers and enablers are patient centred, clinician centred or structural in the Results would be very helpful for the reader, and is recommended. 

3. Related to the reviewer's comments on references, recent article from Parfitt et al, BMC Open Ophthalmol 2019 may be of interest.

4. Minor comment: In the Supplementary Tables, focus groups are indicated with the city name only; could the state please be included and if metropolitan or regional (this is in the manuscript but is useful for the reader to have with the Tables). 

We would appreciate receiving your revised manuscript by Feb 24 2020 11:59PM. To enhance the reproducibility of your results, we recommend that if applicable you deposit your laboratory protocols in protocols.io, where a protocol can be assigned its own identifier (DOI) such that it can be cited independently in the future. For instructions see: http://journals.plos.org/plosone/s/submission-guidelines#loc-laboratory-protocols

We look forward to receiving your revised manuscript.

Kind regards,

Michele Madigan

Academic Editor

PLOS ONE

Journal Requirements:

Reviewers' comments:

Reviewer's Responses to Questions

**Comments to the Author**

1. Is the manuscript technically sound, and do the data support the conclusions?

Reviewer #1: Partly

2. Has the statistical analysis been performed appropriately and rigorously? 

Reviewer #1: Yes

3. Have the authors made all data underlying the findings in their manuscript fully available?

Reviewer #1: No

4. Is the manuscript presented in an intelligible fashion and written in standard English?

Reviewer #1: Yes

5. Review Comments to the Author

Reviewer #1: I enjoyed reading this very well written manuscript on eyecare clinicians’ views about AMD care in Australia. Qualitative research methodology is an important means of gaining a deeper understanding of a phenomenon or subject area than would be possible by using quantitative methods alone, and the results and interpretations presented here certainly benefit from this. Together with the work on patient perspectives that the authors mention is in preparation, this work should help with building useful resources for those looking to improve AMD care in Australia. However, there are a few issues I believe need addressing, these are outlined below.

Crucially, the interview topic guide is heavily weighted towards barriers rather than enablers. For example, the first question listed under the ‘AMD patient journey’ is ‘From your perspective, what are some of the factors preventing people with AMD from accessing AMD care?’ and under ‘Communication with AMD patient’ is ‘From your perspective, what are some of factors preventing people with AMD from following the advice given to them by their practitioner?’. Where questions do ask about enablers they are worded from a more negative perspective, i.e. ‘What could be done to enable better access?’ and ‘What more could be done to help practitioners give the right advice?’, rather than (for example), ‘From your perspective, what factors encourage practitioners to give the right advice’? This, coupled with the fact that not all focus groups were asked about enablers, has meant that results are heavily biased towards barriers.

In addition:

- Barriers and enablers were categorised as being patient centred/clinician centred/structural, yet in the results tables, there is no indication as to how each barrier/enabler listed was categorised. This would be a useful addition, particularly for the clinical reader who may be looking at ways to improve their practice based on results presented.

- On page 24, line 544, the authors refer to ‘new “fast-track” referral pathways’ for wet AMD in the UK. This type of pathway has been around for over 10 years (e.g. Menon, G., Walters, G. New paradigms in the treatment of wet AMD: the impact of anti-VEGF therapy. Eye 23, S1–S7 (2009) doi:10.1038/eye.2009.13) so I am not convinced it could be called ‘new’.

- The paper on healthcare experiences of patients with AMD in the UK (Mitchell et al., 2002; reference 52) discussed under ‘Communication’ on pages 27 and 28 is now dated as the survey was repeated in 2013. For the up-to-date paper, see Boxell EM, Amoaku WM, Bradley C. Healthcare experiences of patients with age-related macular degeneration: have things improved? Cross-sectional survey responses of Macular Society members in 2013 compared with 1999. BMJ Open 2017;7:e012790. doi: 10.1136/bmjopen-2016-012790.

- See also the following for further relevant patient perspectives on patient-clinician communication in dry AMD:

Taylor, D.J., Jones, L., Binns, A.M. et al. ‘You’ve got dry macular degeneration, end of story’: a qualitative study into the experience of living with non-neovascular age-related macular degeneration. Eye (2019) doi:10.1038/s41433-019-0445-8

Carlton, J., Barnes, S. and Haywood, A., 2019. Patient Perspectives in Geographic Atrophy (GA): Exploratory Qualitative Research to Understand the Impact of GA for Patients and Their Families. British and Irish Orthoptic Journal, 15(1), pp.133–141. DOI: http://doi.org/10.22599/bioj.137

6. PLOS authors have the option to publish the peer review history of their article (what does this mean?). If published, this will include your full peer review and any attached files.

Reviewer #1: No

---

## [Author Response · Author response to Decision Letter 0]

13 Jan 2020

Thank you for your positive comments on this research. The following sentence was added to the list of limitations in the discussion section on page 29: “Less than half of the focus groups (three of seven) asked about enablers; the interview topic guide was also heavily weighted towards barriers. This is likely to have biased the results towards a focus on barriers.” A new column was added to each of the 4 relevant supplementary tables (2, 3, 4, and 5). Category themes were then assigned to one or more of the categories of influence and are listed in the additional column in these tables. The following sentence was added to the discussion on page 24: “In line with the findings of this study, a combination of service/healthcare professional-related factors (e.g. inability to get an early appointment, AMD not appropriately detect or labelled) and patient-related factors (e.g. lack of knowledge of AMD symptoms and/or risks) have been cited as common reasons for delay in diagnosis of AMD [40].” The state and whether it is metropolitan or regional has been added to the notes for Supplementary Tables 2, 3, 4, and 5. The following sentence was added to the list of limitations in the discussion section on page 29: “Less than half of the focus groups (three of seven) asked about enablers; the interview topic guide was also heavily weighted towards barriers. This is likely to have biased the results towards a focus on barriers.” A new column was added to each of the 4 relevant supplementary tables (2, 3, 4, and 5). Category themes were then assigned to one or more of the categories of influence and are listed in the additional column in these tables. The word “new” has been removed from the sentence in line 544. The section headed "Communications" has been updated to include these references and now reads as follows on page 28: “Repeated surveys of Macular Disease Society members in the UK highlight significant dissatisfaction with their consultation in people with AMD with 24.2% of people with AMD dissatisfied with their consultation in 2013 [51,52]. The most common reason for dissatisfaction with diagnostic consultation remains poor specialist’s attitudes (often described as dismissive, patronising, brusque, unfeeling, uninterested, etc. – 47.8%) [51,52]. Over half of people with AMD indicated that they do not receive enough information about nutrition and lifestyle choices [53]. Many studies have highlighted the negative impact of an AMD diagnosis on individuals’ emotional wellbeing and the inability of eyecare professionals to provide the type of information and support that is required by people with AMD [54.55]. ……”

---

## [Editor Report · Decision Letter 1]

27 Jan 2020

A qualitative exploration of Australian eyecare professional perspectives on Age-Related Macular Degeneration (AMD) care

PONE-D-19-30557R1

Dear Dr. Jalbert,

We are pleased to inform you that your manuscript has been judged scientifically suitable for publication and will be formally accepted for publication once it complies with all outstanding technical requirements. Please also check that the latest version of Supplementary Information is included. 

With kind regards,

Michele Madigan

Academic Editor

PLOS ONE
---

## [Editor Report · Acceptance letter]

3 Feb 2020

PONE-D-19-30557R1 

A qualitative exploration of Australian eyecare professional perspectives on Age-Related Macular Degeneration (AMD) care 

Dear Dr. Jalbert:

I am pleased to inform you that your manuscript has been deemed suitable for publication in PLOS ONE. Congratulations! Your manuscript is now with our production department. 

With kind regards,

on behalf of

Dr. Michele Madigan 

Academic Editor

PLOS ONE